# Clinical, Biochemical, and Psychological Predictors of Metabolic Syndrome in Climacteric Women

**DOI:** 10.3390/healthcare13243214

**Published:** 2025-12-08

**Authors:** Mauricio Sánchez-Barajas, Marysol García-Pérez, Teodoro Córdova-Fraga, María-Raquel Huerta-Franco

**Affiliations:** 1Department of Internal Medicine, Mexican Social Security Institute (IMSS), León 37320, Guanajuato, Mexico; msanchezb@imss.gob.mx; 2Faculty of Engineering and Technologies, Universidad La Salle Bajío, León 37150, Guanajuato, Mexico; mgarciap@lasallebajio.edu.mx; 3Department of Engineering Physics, Division of Sciences and Engineering, University of Guanajuato, Campus León, León 37150, Guanajuato, Mexico; tcordova@ugto.mx; 4Department of Applied Sciences to Work, Division of Health Sciences, Campus Leon, University of Guanajuato, León 37670, Guanajuato, Mexico

**Keywords:** climacteric, menopause, metabolic syndrome, perceived stress, psychological symptoms

## Abstract

**Highlights:**

Perceived stress, systemic arterial hypertension, Type 2 diabetes mellitus, and elevated fasting glucose independently predicted metabolic syndrome (MS) in climacteric women. Early postmenopausal women presented higher total and HDL-cholesterol levels and greater perceived stress than women in other menopausal stages.

**What are the main findings?**

**What is the implication of the main finding?**

**Abstract:**

**Background/Objectives:** To identify clinical, biochemical, and psychological factors associated with metabolic syndrome (MS) in climacteric women and to determine independent predictors of MS across menopausal stages. **Methods**: A cross-sectional study was conducted in 225 women (perimenopausal, *n* = 75; early postmenopausal, *n* = 75; late postmenopausal, *n* = 75). Anthropometry, clinical history, and fasting laboratory tests were obtained. Psychological measures included perceived stress (Perceived Stress Scale, PSS), anxiety symptoms (Short Health Anxiety Inventory, SHAI-18), and depressive symptoms (Hamilton–Bech–Rafaelsen Scale). **Results**: Perimenopausal women had higher BMI than both postmenopausal groups (33 ± 5 vs. 30 ± 5 and 29 ± 4 kg/m^2^; *F* = 13.39, *p* < 0.001). Waist/hip ratio showed a modest group effect (*F* = 6.34, *p* = 0.002), being higher in perimenopause versus late postmenopause (*p* = 0.001). Significant group differences were observed in lipid and glucose profiles across menopausal stages. Total cholesterol (*F* = 4.86, *p* = 0.009), HDL cholesterol (*F* = 7.12, *p* = 0.001), and non-HDL cholesterol (*F* = 8.13, *p* < 0.001) differed significantly, as confirmed by post hoc Tukey HSD tests, with higher total and non-HDL cholesterol levels in early and late postmenopausal women compared with the perimenopausal group, and higher HDL cholesterol levels in early postmenopausal women compared with the perimenopausal group. Fasting glucose showed a significant difference (*H* = 9.89, *p* = 0.007, Kruskal–Wallis test), with higher median levels in perimenopausal (127 mg/dL) than in early postmenopausal women (97 mg/dL, *p* = 0.003, Mann–Whitney U). Perceived stress was highest in early postmenopause (61.3%) compared with late postmenopause (48.0%) and perimenopause (34.7%), χ^2^ = 10.68, *p* = 0.0048, while anxiety and depression did not differ. Logistic regression analyses identified perceived stress and depressive symptoms as significant predictors of metabolic syndrome under different diagnostic definitions. Higher perceived stress was inversely associated in the psychological model (aOR = 0.62; 95% CI 0.43–0.88; *p* = 0.008) but positively related in the clinical model including fasting glucose and blood pressure (aOR = 1.54; 95% CI 1.07–2.22; *p* = 0.021). In the combined model, both fasting glucose and perceived stress remained independent predictors (*p* < 0.05), under-scoring the contribution of psychological factors to metabolic risk. **Conclusions:** Among climacteric women, perceived stress and cardiometabolic factors (systemic arterial hypertension, Type 2 DM, and elevated fasting glucose) are independent predictors of metabolic syndrome. Early identification and integrated management of stress and metabolic risks may help reduce the burden of metabolic syndrome across menopausal stages.

## 1. Introduction

Menopause represents a natural and inevitable biological transition in a woman’s life, characterized by the permanent cessation of menstruation and the end of reproductive potential. This period is part of the broader climacteric phase, which includes the transition from reproductive to non-reproductive stages. According to the Stages of Reproductive Aging Workshop (STRAW + 10) criteria, the menopausal transition (perimenopause) and postmenopause may extend over several years and involve profound hormonal and metabolic alterations [1,2,3].

During the climacteric period, the gradual decline in ovarian estrogen production is associated with physiological and psychological symptoms, including vasomotor disturbances, mood disorders, and sleep disruptions. In parallel, this hormonal decline contributes to the development or exacerbation of metabolic abnormalities that characterize metabolic syndrome (MS), such as visceral obesity, dyslipidemia, systemic arterial hypertension, and impaired glucose metabolism [4,5,6].

MS is recognized as a major risk factor for cardiovascular disease, which is the leading cause of morbidity and mortality in postmenopausal women [7,8]. The prevalence of MS increases with age and is significantly higher in postmenopausal women compared with premenopausal or perimenopausal counterparts [9,10,11]. Recent epidemiological studies conducted after 2020 have reported global prevalence rates ranging from 30% to 55% in postmenopausal populations across Latin America, Europe, and Asia [12,13].

Emerging international evidence has further highlighted metabolic and psychosocial determinants of health in midlife women. Alterations such as elevated serum alanine aminotransferase, uric acid, and the triglyceride-to-HDL cholesterol ratio (TG/HDL-C) have been shown to distinguish metabolically unhealthy from metabolically healthy obesity phenotypes, suggesting early hepatic–metabolic vulnerability not fully captured by traditional MS criteria [14,15,16]. Additionally, reproductive history markers (including age at menarche, parity, and breastfeeding patterns) have been associated with obesity and obesity-related hypertension in later life, underscoring the long-term cardiometabolic implications of female reproductive trajectories [10]. Moreover, women with a history of miscarriage report poorer sleep quality, more frequent early awakenings, and greater mood disturbances in midlife, reflecting the interplay between reproductive experiences and psychosocial well-being [17]. Integrating these metabolic, reproductive, and emotional dimensions emphasizes the need for studies that jointly assess these determinants, particularly in understudied Latin American populations.

Although numerous investigations have described the metabolic and cardiovascular consequences of menopause, relatively few have explored the co-occurrence of psychological factors such as anxiety, depressive mood, and perceived stress in women with MS, particularly across distinct menopausal stages [12,13]. Growing evidence suggests that psychological stress acts as an additional physiological burden, influencing neuroendocrine and inflammatory pathways that may amplify metabolic risk and cardiovascular vulnerability [12]. This interaction underscores the importance of integrative approaches that jointly evaluate metabolic, hormonal, and psychosocial dimensions in climacteric women [7,8,12].

Given these scientific antecedents, the objective of the present study was to compare clinical, biochemical, and psychological symptom profiles in women at different stages of the climacteric (perimenopause, early postmenopause, and late postmenopause). We hypothesized that MS-related features and psychological disturbances are more prevalent and severe during postmenopause and may act as independent predictors of metabolic syndrome in this population.

## 2. Materials and Methods

Study Reporting. This cross-sectional observational study was designed and reported following the recommendations of the STROBE Statement (Strengthening the Reporting of Observational Studies in Epidemiology) to ensure transparency, quality, and completeness in the description of methods and results. Adherence to the 22-item checklist was applied, covering key aspects such as study design, participant selection, variable definition, data handling, statistical analysis, and clear presentation of findings.

Study Design and Participants. This cross-sectional study was conducted at the Department of Internal Medicine, Clinic 21, Mexican Social Security Institute (IMSS), León, Guanajuato, Mexico. All women attending medical consultation were screened, and those diagnosed with metabolic syndrome were invited to participate after receiving a full explanation of the study objectives and signing a written informed consent form. A total of 225 women aged 40–60 years were enrolled and categorized into three groups based on menopausal status: perimenopausal (PERI, *n* = 75), early postmenopausal (early PM, *n* = 75), and late postmenopausal (late PM, *n* = 75). Classification followed the World Health Organization (WHO) criteria and the Stages of Reproductive Aging Workshop (STRAW + 10) guidelines [1,2,3,4,5]. In accordance with a secondary-tertiary healthcare setting (a specialized hospital environment providing both referred and advanced care), participant recruitment and assessments were conducted as part of routine clinical practice.

Sample Size Calculation. The required sample size (*n* = 225; 75 participants per group) was estimated using G*Power 3.1 for a one-way ANOVA with three groups, assuming a medium effect size (f = 0.25), α = 0.05, and power = 0.90 [18]. The minimum sample required was 207 participants; thus, 225 were recruited to compensate for potential missing data. This sample provides >90% power to detect inter-group differences in metabolic and psychological parameters.

Inclusion and Exclusion Criteria. Data were obtained from standardized clinical, biochemical, and psychological assessments performed during a single visit. Women with regular menstrual cycles but subtle reproductive-hormonal changes were classified as PERI. Early PM included women with 1–5 years since last menstruation, whereas late PM included women with more than 5 years postmenopause. The cutoff of five years after the final menstrual period was selected to define the transition between early and late postmenopause. This criterion was based on local clinical practice guidelines used at our institution and allowed for a more balanced distribution of participants across menopausal stages while remaining consistent with the physiological framework proposed by the STRAW + 10 criteria. Classification was based on clinical symptoms and menstrual history and was confirmed by medical evaluation in accordance with STRAW + 10 guidelines [1]. Exclusion criteria were pregnancy, breastfeeding, acute or chronic infectious, metabolic, or cardiovascular diseases, and recent use of anxiolytics, antidepressants, analgesics, vitamins, antibiotics, or hormone replacement therapy. Women with hysterectomy were eligible if their hormonal profile indicated postmenopausal status, defined as serum follicle-stimulating hormone (FSH) ≥ 40 IU/L and estradiol ≤ 30 pg/mL. These biochemical criteria were applied because menstrual cycle data were unavailable for this subgroup. In addition, all participants were required to be at least 45 years old and to have documented absence of ovarian function according to their clinical records. All participants met the diagnostic criteria for metabolic syndrome, which were consistent with the definitions commonly applied in recent research on menopause and metabolic syndrome [6]. (See explanation below).

Participants completed validated questionnaires assessing physical symptoms (e.g., vasomotor symptoms, sleep alterations) and psychological parameters including perceived stress, anxiety, and depression.

Perceived stress was measured using Cohen’s Perceived Stress Scale (PSS-14), consisting of 14 items rated from 0 (“never”) to 4 (“very often”). The Spanish version validated in Mexican adults by González-Ramírez et al. showed an α = 0.83 and strong construct validity [19].

Anxiety was assessed using the Short Health Anxiety Inventory (SHAI-18), an 18-item self-report questionnaire scored from 0 to 3 per item. The Spanish version validated by Arnáez et al. demonstrated good internal consistency (α = 0.80) and factorial validity in Mexican populations [20].

Depressive symptoms were assessed using the Hamilton–Bech–Rafaelsen Depression Scale (HBRDS), a 6-item clinician-rated instrument (rated from 0 to 4 each item), with excellent internal consistency (Cronbach’s α = 0.89) and criterion validity against the Hamilton–Bech–Rafaelsen Depression Rating Scale. This instrument has been previously adapted and applied in Mexican menopausal women by Huerta-Franco et al., demonstrating its practical suitability for use in this population [21].

All questionnaires were administered in their validated Spanish versions, following standard administration guidelines and prior psychometric validation in Mexican or Latin American samples [19,20,21].

Anthropometric and Laboratory Measurements. Anthropometric parameters including body weight, height, waist and hip circumferences were obtained by trained personnel according to the International Society for the Advancement of Kinanthropometry (ISAK) standards [21]. All measurements were performed using calibrated instruments and standardized protocols to ensure reliability and inter-observer consistency. Body composition was evaluated using both skinfold thickness and bioelectrical impedance analysis (BIA). Given its superior precision and reproducibility in clinical contexts, body fat percentage was calculated from BIA-derived values [21].

Fasting venous blood samples were obtained between 07:00 and 09:00 h after a 10–12 h overnight fast. Laboratory analyses included glucose, total cholesterol, HDL cholesterol, non-HDL cholesterol, and triglycerides, measured by enzymatic-colorimetric methods (Roche Diagnostics GmbH, Mannheim, Germany). Blood pressure (BP) was recorded in triplicate after 10 min of rest, using a digital sphygmomanometer (Omron HEM-907; Omron Healthcare Inc., Lake Forest, IL, USA), and the mean value of the two lowest readings was used.

Metabolic syndrome (MS) was defined according to the National Cholesterol Education Program Adult Treatment Panel III (NCEP-ATP III) criteria as the presence of three or more of the following components: waist circumference > 88 cm; triglycerides ≥ 1.7 mmol/L (150 mg/dL); HDL cholesterol < 1.29 mmol/L (50 mg/dL); blood pressure ≥ 130/85 mmHg or treatment for systemic arterial hypertension; fasting plasma glucose ≥ 6.1 mmol/L (110 mg/dL). These definitions were cross checked with the World Health Organization (WHO, 1999) and International Diabetes Federation (IDF) criteria to ensure diagnostic consistency [22].

### Statistical Analysis

Data were analyzed using IBM SPSS Statistics version 21 (IBM Corp., Armonk, NY, USA) and Python version 3.10 (pandas and statsmodels; Python Software Foundation, Wilmington, DE, USA). The normality of quantitative variables was assessed using the Kolmogorov–Smirnov test. To evaluate group differences in laboratory variables across menopausal stages, a one-way ANOVA with Tukey’s HDS post hoc test was applied for normally distributed biochemical parameters (total cholesterol, HDL cholesterol, and non-HDL cholesterol). For fasting glucose, which did not meet normality assumptions, non-parametric analyses were conducted using the Kruskal–Wallis test followed by pairwise Mann–Whitney U tests to identify specific group differences.

Multivariate Logistic Models Identifying Clinical and Psychological Predictors of Metabolic Syndrome. Three derived models were defined: SM1 combined waist circumference (>88 cm), triglycerides (≥150 mg/dL), and HDL cholesterol (<50 mg/dL); SM2 included blood pressure (≥130/85 mmHg or a diagnosis of hypertension), triglycerides (≥150 mg/dL), and fasting glucose (≥110 mg/dL); and SM3 comprised waist circumference (>88 cm), HDL cholesterol (<50 mg/dL), and fasting glucose (≥110 mg/dL). Each variable was coded as 1 = meets all three criteria and 0 = otherwise, allowing for comparative logistic regression analyses. Predictor variables included perceived stress, anxiety, and depression obtained from validated questionnaires. Continuous psychological scores were standardized (z-scores) to facilitate interpretation per one standard deviation (SD) increase. Clinical predictors included systemic arterial hypertension, type 2 diabetes mellitus (DM2), and elevated fasting glucose (≥110 mg/dL). A binary variable for perceived stress (1 = stress present) was used in sensitivity analyses. Logistic regression models were fitted using SM1, SM2, and SM3 as dependent outcomes, and clinical and/or psychological variables as independent predictors. Results are reported as adjusted odds ratios (aOR) with 95% confidence intervals (95% CI). Multicollinearity was evaluated using variance inflation factors (VIF), with values >10 considered problematic. Model performance was compared using Akaike’s Information Criterion (AIC) and pseudo-R^2^ as indicators of model fit. All analyses were two-tailed, and a *p* value < 0.05 was considered statistically significant.

Ethical Considerations. The study protocol was approved by the IMSS Ethics Committee (approval no. 2019-1008). Written informed consent was obtained from all participants in accordance with the Declaration of Helsinki. All procedures adhered to national and institutional research ethics standards and complied with the Mexican Official Standard NOM-012-SSA3-2012 for research involving human subjects. Participant confidentiality and data protection were guaranteed in accordance with IMSS institutional policies and national data protection regulations. All personal information was anonymized and stored in secure institutional databases accessible only to the authorized research team.

## 3. Results

### 3.1. Participant Characteristics

A total of 225 climacteric women were evaluated, equally distributed among the three groups: PERI (*n* = 75), early PM (*n* = 75), and late PM (*n* = 75). The mean ages were 44.1 ± 5.5 years in PERI, 52.6 ± 3.9 years in early PM, and 57.7 ± 5.4 years in late PM, with highly significant differences among groups (ANOVA F = 142.33, *p* < 0.001); post hoc analyses confirmed a stepwise increase (PERI < early PM < late PM, all *p* < 0.001). Detailed characteristics are provided in Table 1.

Body mass index (BMI) indicated that PERI women had significatively higher BMI (mean BMI 32.7 ± 4.6 kg/m^2^) than both postmenopausal groups (29.6 ± 4.6 and 29.2 ± 4.4 kg/m^2^, respectively). These differences were significant (*F* = 13.39, *p* < 0.001), with PERI > early PM and PERI > late PM (both *p* < 0.001), while no difference was observed between the postmenopausal groups (*p* = 0.59).

Waist-to-hip ratio (WHR) suggested an android fat distribution across all groups (0.91 ± 0.05; 0.89 ± 0.06; 0.87 ± 0.09), with ANOVA indicating a modest but significant group effect (*F* = 6.34, *p* = 0.002). Post hoc testing revealed higher WHR in PERI versus late PM (*p* = 0.001), whereas comparisons between PERI vs. early PM and early PM vs. late PM did not reach significance (*p* = 0.153 and *p* = 0.08, respectively).

Regarding categorical variables, education level did not show differences among groups (χ^2^ = 11.08, *p* = 0.09), with a greater proportion of bachelor-professional education in early PM and more illiteracy in late PM. Occupation did not differ significantly (χ^2^ = 2.72, *p* = 0.26), as the distribution of housewives and workers was similar across groups.

#### 3.1.1. Biochemical and Clinical Parameters

Laboratory assessments revealed that approximately 50% of PERI women exhibited elevated fasting glucose levels. Both early and late PM groups demonstrated a higher prevalence of hypercholesterolemia, whereas systemic arterial hypertension was more frequent in the late PM group. Type 2 DM was prevalent in all groups, with the highest proportion observed in late PM women (82.7%). Elevated triglyceride levels were also frequent in this group. Significant group differences were found for fasting glucose (χ^2^ = 14.17, df = 2, *p* = 0.0008), total cholesterol (χ^2^ = 9.62, df = 4, *p* = 0.047), HDL (χ^2^ = 13.99, df = 4, *p* = 0.007), and non-HDL cholesterol (χ^2^ = 17.40, df = 6, *p* = 0.008); no significant differences were observed for triglycerides (χ^2^ = 7.33, df = 6, *p* = 0.291), systemic arterial systemic arterial hypertension (χ^2^ = 4.07, df = 2, *p* = 0.131), or Type 2 DM (χ^2^ = 1.43, df = 2, *p* = 0.489). Categorized biochemical and clinical parameters by group are summarized in Table 2.

#### 3.1.2. Group Differences in Glucose and Lipid Profiles

To assess metabolic variations across menopausal stages, one-way ANOVA and non-parametric analyses were performed Appendix A. ANOVA revealed significant group effects for total cholesterol (*F* = 4.86, *p* = 0.009), HDL cholesterol (*F* = 7.12, *p* = 0.001), and non-HDL cholesterol (*F* = 8.13, *p* < 0.001). The mean (±SD) values were 181.2 ± 32.8, 196.9 ± 31.5, and 195.4 ± 37.3 mg/dL for total cholesterol; 48.7 ± 8.5, 55.1 ± 11.3, and 50.9 ± 11.2 mg/dL for HDL cholesterol; and 93.4 ± 27.3, 107.9 ± 30.6, and 113.5 ± 35.9 mg/dL for non-HDL cholesterol, respectively, in perimenopausal, early postmenopausal, and late postmenopausal women. Post hoc Tukey HSD comparisons indicated that total and non-HDL cholesterol were significantly higher in early and late postmenopausal women compared with perimenopausal women (*p* < 0.05), whereas HDL cholesterol was significantly higher in early postmenopause compared with perimenopause Appendix A.

Because fasting glucose did not meet normality assumptions, a Kruskal–Wallis test was conducted, revealing significant differences among groups (*H* = 9.89, *p* = 0.007). Median glucose values were 127 mg/dL for perimenopausal, 97 mg/dL for early postmenopausal, and 105 mg/dL for late postmenopausal women. Pairwise Mann–Whitney U tests confirmed that perimenopausal women exhibited significantly higher glucose concentrations compared with early postmenopausal women (*p* = 0.003), while no significant differences were observed between early and late postmenopausal stages (*p* > 0.05). Appendix A.

#### 3.1.3. Psychological Symptoms and Stress Perception

Perceived stress was highest in early PM women (61.3%), followed by late PM (48%) and PERI (34.7%) (χ^2^ = 10.68, df = 2, *p* = 0.0048). Mild depressive symptoms were highly prevalent in all groups, with late PM women showing the highest frequency (85.3%) (χ^2^ = 3.02, df = 4, *p* = 0.55). Moderate anxiety was observed in 40% of early PM, 32% of late PM, and 28% of PERI participants; no significant group differences were found (χ^2^ = 2.55, df = 4, *p* = 0.636). Group-wise distributions of psychological outcomes are presented in Table 3.

#### 3.1.4. Predictors of Metabolic Syndrome

Although all participants met the diagnostic criteria for metabolic syndrome, logistic regression models were conducted to explore how different combinations of diagnostic components and psychological variables interacted within this population.

In the psychological model (SM1, including waist circumference, triglycerides, and HDL cholesterol), perceived stress and depressive symptoms were significantly associated with metabolic syndrome. Higher perceived stress was inversely related to SM1 (aOR = 0.618; 95% CI 0.433–0.882; *p* = 0.008), whereas depressive symptoms were positively associated (aOR = 1.479; 95% CI 1.005–2.175; *p* = 0.047). Anxiety was not significantly related (aOR = 0.985; 95% CI 0.664–1.463; *p* = 0.942).

In contrast, when the alternative definition SM2 (blood pressure, triglycerides, and fasting glucose) was used, the direction of the association for perceived stress shifted. In this model, higher perceived stress was directly associated with metabolic syndrome (aOR = 1.542; 95% CI 1.072–2.219; *p* = 0.021), consistent with its physiological role in cardiometabolic dysregulation. Anxiety (aOR = 1.138; 95% CI 0.784–1.653; *p* = 0.493) and depression (aOR = 1.081; 95% CI 0.768–1.521; *p* = 0.647) remained non-significant predictors. Variance inflation factors (VIFs < 3) indicated no problematic multicollinearity.

When the SM3 definition (waist circumference, HDL cholesterol, and fasting glucose) was applied, results were consistent with previous models. In the clinical model, elevated fasting glucose (aOR = 2.864; 95% CI 1.379–5.944; *p* = 0.005) and type 2 diabetes (aOR = 2.102; 95% CI 1.082–4.083; *p* = 0.028) were the strongest independent predictors, whereas hypertension showed no significant association (aOR = 1.124; 95% CI 0.672–1.880; *p* = 0.651).

In the combined model including perceived stress, both fasting glucose (aOR = 2.711; 95% CI 1.304–5.631; *p* = 0.007) and perceived stress levels (aOR = 1.478; 95% CI 1.016–2.150; *p* = 0.042) remained significant predictors of metabolic syndrome. These results suggest that psychological stress exerts an independent influence on metabolic risk, even after adjustment for clinical factors (see Figure 1). All variance inflation factors remained below 3, confirming the absence of multicollinearity among predictors Appendix A.

## 4. Discussion

Perimenopausal women showed higher BMI (values on average in the obesity range), while both early and late postmenopausal groups remained in the overweight range [23,24]. These findings suggest that perimenopause represents a critical period during which hormonal changes begin to influence metabolic regulation [23]. The modest but significant difference in WHR between PERI and late PM supports the notion that central adiposity is already established during the menopausal transition [24,25]. Significant group differences for fasting glucose, total cholesterol, HDL, and non-HDL cholesterol indicate that metabolic disruption evolves differently across climacteric stages [26,27,28,29].

The high proportion of PERI women with elevated fasting glucose suggests that glucose alterations appear even before the final menstrual period [28]. In contrast, both early and late postmenopausal groups showed a higher prevalence of hypercholesterolemia, consistent with the loss of estrogen’s protective role in lipid metabolism [23,29].

Although triglycerides, systemic arterial hypertension, and Type 2 DM did not differ significantly among groups, their overall prevalence highlights the cumulative burden of cardiometabolic risk factors during the climacteric [5,6,7,8,9,10]. Compared with population patterns in which younger adults tend to report higher perceived stress than older adults, our cohort showed a stage-specific peak of perceived stress in early postmenopause, exceeding levels in the (younger) perimenopausal group [16,30,31,32,33]. This suggests that menopausal stage drivers (sleep disruption, vasomotor symptoms, cardiometabolic comorbidity) may override the usual age gradient in perceived stress [33,34,35].

Based on comparable populations, reference values for the Perceived Stress Scale (PSS) suggest that the proportion of women with high perceived stress in early postmenopause in this study likely exceeds population-level expectations [33]. Our results partially align with international evidence on mood disorders [35]. Large longitudinal cohorts (SWAN) report higher odds of major depression in peri- and early postmenopause compared with premenopause, independent of prior depression and vasomotor symptoms [34]. A recent global meta-analysis also highlights the high burden of depression in menopausal women [32]. In our data, anxiety and depressive symptoms did not differ significantly by climacteric stage, while perceived stress did. This may reflect methodological differences, including the type of psychological instruments applied, as well as contextual factors such as sociodemographic stressors known to influence perceived stress [36,37].

These findings converge with global literature in highlighting midlife mental-health vulnerability, while adding nuance by demonstrating a distinct early postmenopausal peak in perceived stress within a cardiometabolic-risk sample [38,39]. Clinically, this supports integrated care: routine stress screening (PSS), management of sleep and vasomotor symptoms, and targeted psychoeducational and behavioral interventions alongside metabolic risk control [34].

Beyond behavioral and psychosocial explanations, the relationship between perceived stress and metabolic risk also involves neuroendocrine and inflammatory mechanisms. Chronic activation of the hypothalamic–pituitary–adrenal (HPA) axis under sustained stress increases circulating cortisol and catecholamines, which stimulate hepatic gluconeogenesis, visceral adiposity, and insulin resistance, thereby promoting metabolic-syndrome components [40,41]. Sympathetic over-activation further elevates blood pressure and impairs endothelial function [42,43,44]. Persistent exposure to cortisol and pro-inflammatory cytokines such as IL-6 and TNF-α enhances adipocyte differentiation and hepatic lipid accumulation, amplifying insulin resistance and dyslipidemia [45,46]. Moreover, the decline in estrogen characteristics of the menopausal transition reduces glucocorticoid-receptor sensitivity and weakens estrogen’s anti-inflammatory and antioxidant effects, thereby exacerbating stress-related metabolic dysregulation [47].

In addition to these neuroendocrine and inflammatory pathways, recent evidence indicates that specific metabolic biomarkers may also contribute to the pathophysiology of metabolic syndrome in midlife women. Elevated serum uric acid levels and an increased triglyceride to HDL cholesterol ratio (TG/HDL-C) have been associated with insulin resistance, abdominal obesity, and cardiovascular risk, particularly during the menopausal transition. Furthermore, the uric acid to HDL cholesterol ratio (UHR) has been identified as a novel indicator of central adiposity and metabolic dysfunction in postmenopausal women. These findings highlight the multifactorial nature of metabolic syndrome, integrating hepatic, lipid, and hormonal alterations beyond traditional cardiometabolic components [14,15,17].

Taken together, these neuroendocrine-immune alterations provide a plausible biological framework linking psychological stress with the clinical and biochemical manifestations of metabolic syndrome in climacteric women.

### Limitations

This study has some limitations that should be considered when interpreting the results. (1) Its cross-sectional design precludes causal inferences between perceived stress, hormonal changes, and metabolic syndrome components. (2) Psychological variables were self-reported, which may introduce recall bias despite using validated instruments. (3) Inflammatory biomarkers (IL-6, TNF-α), serum uric acid, and glycated hemoglobin (HbA1c) were not measured, limiting the assessment of metabolic and inflammatory predictors. Climacteric symptoms were collected descriptively but excluded from regression analyses due to incomplete data. (4) As participants were recruited from a single healthcare center, generalizability may be limited.

Despite these limitations, the study provides valuable evidence on the clinical, biochemical, and psychological correlates of metabolic risk during the climacteric transition in Mexican women.

## 5. Conclusions

Metabolic and psychological vulnerabilities vary across menopausal stages in women with metabolic syndrome. Perimenopausal women exhibited greater obesity and elevated fasting glucose, whereas postmenopausal women showed more frequent dyslipidemia and higher perceived stress. Perceived stress, systemic arterial hypertension, Type 2 diabetes mellitus, and elevated fasting glucose were significant predictors of metabolic syndrome, highlighting the interaction between clinical and psychosocial determinants of health.

These findings emphasize the need for integrative care strategies that combine metabolic monitoring with attention to emotional well-being in climacteric women. In clinical practice, stress-management and lifestyle-based interventions (including relaxation or mindfulness techniques, cognitive-behavioral counseling, balanced nutrition, and regular physical activity) may help mitigate stress-related metabolic alterations. Moreover, early metabolic screening during perimenopause and early postmenopause should be incorporated into midlife preventive programs to reduce long-term cardiometabolic risk. Future longitudinal research is required to clarify causal mechanisms and to determine the role of autonomic-nervous-system dysregulation in the development of metabolic syndrome.

## Figures and Tables

**Figure 1 healthcare-13-03214-f001:**
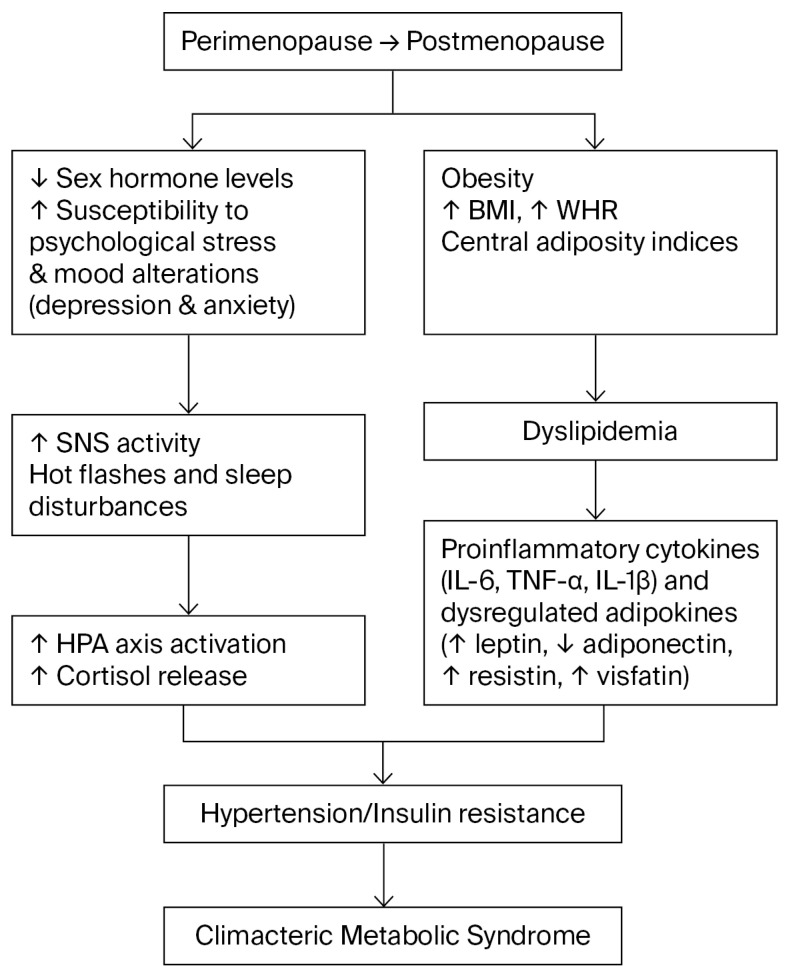
Conceptual flow diagram summarizing the interrelationships among neuroendocrine, psychological, and metabolic factors associated with metabolic syndrome during the climacteric transition. The abrupt decline in sex hormone levels increases vulnerability to psychological stress, anxiety, and depressive symptoms. This heightened stress reactivity involves overactivation of the sympathetic nervous system (SNS) and hypothalamic–pituitary–adrenal (HPA) axis, leading to elevated cortisol levels, central obesity, and systemic arterial hypertension. The menopausal transition is also accompanied by increased abdominal adiposity, dyslipidemia, and a proinflammatory state characterized by altered cytokines and adipokines, which together contribute to metabolic dysregulation in midlife women.

**Table 1 healthcare-13-03214-t001:** Comparative analysis of general characteristics among the three groups of women [mean ± SD; categorical variables as *n* (%)].

Variable	PERIM (*n* = 75)	Early PM (*n* = 75)	Late PM (*n* = 75)	Total (*n* = 225)	Test (Global)
Age (years)	44 ± 5	52 ± 3	57 ± 5	51 ± 7	F = 142.33, *p* < 0.001
BMI (kg/m^2^)	32.7 ± 4.6	29.6 ± 4.6	29.2 ± 4.4	30.5 ± 4.8	F = 13.39, *p* < 0.001
Waist/hip ratio	0.91 ± 0.05	0.89 ± 0.06	0.87 ± 0.09	0.89 ± 0.07	F = 6.34, *p* = 0.002
	No. (%)	No. (%)	No. (%)	No. (%)	
Education level					χ^2^ = 11.08, *p* = 0.09
Illiterate	3 (4.0)	2 (2.7)	8 (10.7)	13 (5.8)	
Middle school	34 (45.3)	24 (32.0)	30 (40.0)	88 (39.1)	
High school	14 (18.7)	11 (14.7)	9 (12.0)	34 (15.1)	
Bachelor/Professional	24 (32.0)	38 (50.7)	28 (37.3)	90 (40.0)	
Occupation					χ^2^ = 2.72, *p* = 0.260
Housewife	49 (65.3)	47 (62.7)	56 (74.7)	152 (67.6)	
Worker	26 (34.7)	28 (37.3)	19 (25.3)	73 (32.4)	

BMI = Body Mass Index, PERIM = Perimenopausal women; Early PM = Early postmenopause (≤5 years since final menstrual period); Late PM = Late postmenopause (>5 years since final menstrual period). Classification according to WHO and STRAW + 10 criteria [1,2,3,4,5]. Continuous variables were compared across groups using one-way analysis of variance with Holm-adjusted post hoc tests, categorical variables with Pearson’s chi-square. Post hoc results: Age: PERIM < Early PM < Late PM (all *p* < 0.001). BMI: PERIM > Early PM (*p* < 0.001), PERIM > Late PM (*p* < 0.001), Early PM vs. Late PM not significant (*p* = 0.590). Waist/hip ratio: PERIM > Late PM (*p* = 0.001); PERIM vs. Early PM (ns, *p* = 0.153); Early PM vs. Late PM (ns, *p* = 0.076).

**Table 2 healthcare-13-03214-t002:** Frequency distribution of categorized biochemical values in the three groups of women (*n* (%)) and chi-square tests.

Variable	Category	PERIM (*n* = 75)	Early PM (*n* = 75)	Late PM (*n* = 75)	χ^2^, *p*-Value
Glucose (mg/dL)	Normal	24 (32.0)	40 (53.3)	33 (44.0)	χ^2^ = 14.2, *p* = 0.001
	High	38 (50.7)	17 (22.7)	17 (22.7)	
Total Chlt (mg/dL)	Low	36 (48.0)	19 (25.3)	24 (32.0)	χ^2^ = 9.62, *p* = 0.047
	Normal	18 (24.0)	22 (29.3)	19 (25.3)	
	High	21 (28.0)	34 (45.3)	32 (42.7)	
HDL-Chlt (mg/dL)	Low	10 (13.3)	6 (8.0)	10 (13.3)	χ^2^ = 14.0, *p* = 0.007
	Normal	56 (74.7)	41 (54.7)	49 (65.3)	
	High	9 (12.0)	28 (37.3)	16 (21.3)	
Non-HDL Chlt (mg/dL)	Normal	67 (89.3)	56 (74.4)	57 (76.0)	χ^2^ = 17.4, *p* = 0.008
	Borderline	8 (10.7)	16 (21.3)	12 (16.0)	
	High risk	0 (0.0)	3 (4.0)	1 (1.3)	
	Very high risk	0 (0.0)	0 (0.0)	5 (6.7)	
Triglycerides (mg/dL)	Normal	27 (36.0)	35 (46.7)	27 (36.0)	χ^2^ = 7.33, *p* = 0.29
	High limit	23 (30.7)	25 (33.3)	21 (28.0)	
	High	24 (32.0)	15 (20.0)	27 (36.0)	
	Very high	1 (1.3)	0 (0.0)	0 (0.0)	
SHT	Negative	34 (45.3)	35 (46.7)	24 (32.0)	χ^2^ = 4.07, *p* = 0.13
	Positive	41 (54.7)	40 (53.3)	51 (68.0)	
T2DM	Negative	16 (21.3)	19 (25.3)	13 (17.3)	χ^2^ = 1.43, *p* = 0.49
	Positive	59 (78.7)	56 (74.7)	62 (82.7)	

PERIM = Perimenopausal women; Early PM = Early postmenopause (≤5 years since final menstrual period); Late PM = Late postmenopause (>5 years since final menstrual period). Classification according to World Health Organization and STRAW + 10 criteria [1,2,3,4,5]. SHT = Systemic arterial hypertension. T2DM = Type 2 Diabetes Mellitus.

**Table 3 healthcare-13-03214-t003:** Symptom profiles in perimenopausal, early postmenopausal, and late postmenopausal women (data are expressed as *n* (%)).

Variable	PERIM (*n* = 75)	Early PM (*n* = 75)	Late PM (*n* = 75)	χ^2^, *p*-Value
Perceived stress (range 0–64)				χ^2^ = 10.68, *p* = 0.005
Do not present stress	49 (65.3)	29 (38.7)	39 (52.0)	
Present stress	26 (34.7)	46 (61.3)	36 (48.0)	
Depressed mood (range 0–26)				χ^2^ = 3.02, *p* = 0.55
Mild	61 (81.3)	63 (84.0)	64 (85.3)	
Moderate	14 (18.7)	10 (13.3)	10 (13.3)	
Severe	0 (0.0)	2 (2.7)	1 (1.3)	
Anxiety (range 0–18)				χ^2^ = 2.55, *p* = 0.636
Mild	41 (54.7)	34 (45.3)	38 (50.7)	
Moderate	21 (28.0)	30 (40.0)	24 (32.0)	
Severe	13 (17.3)	11 (14.7)	13 (17.3)	

PERIM = Perimenopausal women; Early PM = Early postmenopause (≤5 years since final menstrual period); Late PM = Late postmenopause (>5 years since final menstrual period). The Hamilton–Bech–Rafaelsen Depression Rating Scale [21] was used for depressive symptoms, an 18-item self-report questionnaire assessed anxiety [20] and perceived stress was measured using Cohen’s Perceived Stress Scale [19].

## Data Availability

The data presented in this study are available in the Appendix A. Additional data supporting the findings of this study are available on reasonable requests from the corresponding author. The data are not publicly available due to privacy and ethical restrictions.

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
