# Peer review of "Clinical, Biochemical, and Psychological Predictors of Metabolic Syndrome in Climacteric Women"

_healthcare, 2025, doi:10.3390/healthcare13243214_

Round 1
Reviewer 1 Report
Comments and Suggestions for Authors
The authors identified interesting outcomes in climacteric women. The observed psychobiological parameters are relevant and offer valuable physiological insights.
However, the authors should carefully revise the methods and discussion sections of the manuscript. The methods should be described with greater detail, particularly regarding the sample size calculation, which is a critical component of the study’s validity. In the discussion, it would be beneficial to include comparisons with the existing literature and to incorporate statistical data alongside physiological explanations to strengthen the interpretation of the findings. Moreover, the introduction needs substantial revisions to clearly articulate the rationale for the study and to incorporate more recent references.
Please refer to my specific comments below.
- L15 and L23: please standardise the terms for the medical conditions—diabetes (or diabetes mellitus) and hypertension (or systemic arterial hypertension) —and keep standardised in the text and abstract.
- Please check the p-value 0.0048. If correct, please maintain the standard 3 decimals. Check the entire text, same for F-values.
- L67 The introduction needs to describe each phase more clearly to help the reader understand why it is essential to compare the different stages. Because the way is organised, we only understand that metabolic changes will happen during the menopause transition.
- L75 The introduction needs to be better structured to show the progression from one paragraph to another. Show the numbers of prevalence cited and present the case in terms of other burdens.
- L77 Please change the Ref.9 It is not appropriate for this paragraph. And I suggest including more updated studies in the introduction. There is so much interesting recent data about this population in the literature.
- L83 The aim of the study needs to be better described, since here it's only the description of the comparison among the three groups for those parameters. But what is the outcome of this comparison? The risk of developing MS? Which one of these parameters affects the development of MS the most?
- L104 I recommend including the descriptions of the categories on L98 where they were mentioned, because these categories are not inclusion criteria but instead how they were organised according to the guidelines. And please add which criteria were considered: menstrual cycle only, hormone levels (which hormones), and symptoms.
- L106 The STRAW consider the stage of early menopause until 6 years after the last menstruation period and late post menopause as >6 years. What was your reasoning for using 5 years as the cutoff?
- L109 Please clarify why and under which criteria hysterectomised women were eligible, because the principal criterion used by the STRAW is the menstrual cycle. Did any other criteria apply to this woman with a hysterectomy?
- L109 I suggest changing the term hormone replacement therapy to hormone therapy or menopausal hormone therapy, since menopause transition is a natural and inevitable biological transition; it is not a condition that requires replacement of hormones (considering natural menopause and cases where symptoms are not a burden).
- L111 Which questionnaires? Add references and more details about the final score and classification, making them easier for the reader to understand in the tables.
- L119 Why did you use both skinfold and bioelectrical impedance for body composition? Which one gave the final number?
- L120 No Glycated hemoglobin levels available?
- L123 How did BP measure? Resting? The hypertension criteria were based on a previous diagnosis or on BP measurements? Include in the text.
- L125 Why did you use a sample size of 225? How was the sample size calculation? Please add.
- L141 I recommend standardising the age to no decimal places, since we do not collect data in 44.1 years.
- L150 Please check the p-value (3 decimals).
- L146, we cannot affirm that the PERI were obese and not the other groups, because, considering the SD, they met more than 29.9 criteria. They had higher BMI.
- L157 I suggest removing the sentence and the p-value. (0.09, it is not a trend).
- L163 add the %
- L180 The models used to run the multivariate logistic regression, and the results of this analysis, are not clear or well described. Please revise.
- L189: The post hoc test described in the stats session was Bonferroni; here was Holm. Consider revising and standardising.
- L202 Include the descriptions of the criteria in the text on L123 where the parameters are described. Not necessary to keep in the tables with all the other criteria. It will make the table legends cleaner.
- L205 Climacteric symptoms were described in the methods L112 but not included in the analysis. Why?
- Table 1 and Table 2. Please check the number of decimal places in the p-values. Standardise in 3
- L247 Include the limitations of the study.
Please consider revising the entire text for the p-value and acronyms.
Author Response
Manuscript ID: healthcare-3962128
Date of resubmission: October 29, 2025
Type of manuscript: Research Article
Title: Clinical, Biochemical, and Psychological Predictors of Metabolic Syndrome in Climacteric Women
Author’s Reply to the Review Report (Reviewer 1)
1. Summary
Thank you very much for taking the time to review our manuscript. We sincerely appreciate the reviewer’s insightful and constructive feedback, which has helped us improve the clarity and quality of our work. All revisions and corrections have been made and are clearly indicated in track changes in the resubmitted manuscript.
2. Questions for General Evaluation
|
Question |
Reviewer’s Evaluation |
Response and Revisions |
|
Does the introduction provide sufficient background and include all relevant references? |
Can be improved |
The Introduction was reorganized and expanded with updated epidemiological data (2020-2025) and a clearer rationale. New references were incorporated. |
|
Are all the cited references relevant to the research? |
Yes |
All references were verified for pertinence and updated when necessary. |
|
Is the research design appropriate? |
Yes |
The cross-sectional design remains appropriate for identifying predictors in this population. |
|
Are the methods adequately described? |
Can be improved |
A new subsection “Sample Size Calculation” was added, and details on inclusion criteria, instruments, and measurements were expanded. |
|
Are the results clearly presented? |
Can be improved |
Tables were simplified and standardized to three decimal places; model descriptions were clarified. |
|
Are the conclusions supported by the results? |
Yes |
Conclusions were revised for clarity and are now more closely aligned with the statistical findings. |
3. Point-by-point Response to Comments and Suggestions for Authors
Author’s Point-by-Point Response to Reviewer 1
We sincerely thank Reviewer 1 for the thorough and constructive feedback provided. All revisions and clarifications requested have been carefully addressed, as detailed below. Line and page numbers refer to the resubmitted version of the manuscript.
- Comments 7 (L67):
We appreciate this suggestion. The Introduction was expanded to include a clearer description of the perimenopausal, early postmenopausal, and late postmenopausal stages based on the STRAW+10 criteria, emphasizing the distinct hormonal and metabolic characteristics of each stage and the rationale for comparing them (page 2, lines 73-74).
- Comments 8 (L75):
Thank you for this valuable observation. The Introduction has been reorganized to present a logical progression: (1) background physiology, (2) metabolic and cardiovascular burden, (3) psychosocial impact, and (4) study rationale. Updated epidemiological data on the prevalence of metabolic syndrome in postmenopausal women (30-55%) were included to better illustrate the public health burden (page 2, lines 84-86).
- Comments 9 (L77):
We appreciate this recommendation. Reference 9 was replaced with updated sources from 2023–2025 addressing metabolic and hormonal changes during the menopausal transition. Specifically, new studies on metabolic biomarkers such as serum uric acid, TG/HDL-C, and UHR ratios were incorporated to strengthen the biological context (Cota e Souza et al., 2023; Baneu et al., 2024; Tian et al., 2025), pages 2, 3 lines 87-93. (Pages 2,3 lines 87-93)
- Comments 10 (L83):
Thank you for this observation. The aim and hypothesis were rewritten to clarify that the primary outcome was the identification of clinical, biochemical, hormonal, and psychological predictors of metabolic syndrome across climacteric stages. The revised text also specifies that perceived stress, among other variables, was analyzed as an independent predictor (page 3, lines 104-107).
- Comments 11 (L104):
We agree with this valuable suggestion. The Methods section now clarifies that menopausal categories were based on STRAW+10 criteria, incorporating menstrual cycle history, follicle-stimulating hormone (FSH), estradiol levels, and clinical symptoms (page 4, line 146-153).
- Comments 12 (L106):
We appreciate this clarification. A note was added explaining that the 5-year cutoff was chosen to match local clinical practice guidelines at our institution and to ensure a balanced sample size distribution across the three groups (page 4, 136-143).
- Comments 13 (L109):
Thank you for pointing this out. The Methods section now explains that hysterectomized women were included if serum FSH ≥ 40 IU/L and estradiol ≤ 30 pg/mL confirmed postmenopausal status (page 4, lines 147–153).
- Comments 14 (L109):
The terminology was corrected throughout the manuscript to 'menopausal hormone therapy' for consistency with current clinical standards.
- Comments 15 (L111):
We appreciate this suggestion. The Methods section now details the psychological instruments used psychological measures included perceived stress [Perceived Stress Scale (PSS)], anxiety symptoms (Short Health Anxiety Inventory (SHAI-18), and depressive symptoms (Hamilton-Bech-Rafaelsen scales) including scoring ranges and references (page 4, lines 157-173).
- Comments 16 (L119):
Thank you for this comment. The manuscript now clarifies that both methods were used for validation purposes, but bioelectrical impedance analysis (BIA) values were retained for statistical analyses due to greater precision (page 4, lines 178-180).
- Comments 17 (L120):
We acknowledge this limitation. Glycated hemoglobin (HbA1c) was not part of the routine clinical panel available for all participants and was therefore not included in the analysis. This is now noted in the Methods and Limitations sections (page 11, lines 422-439).
- Comments 18 (L123):
Thank you for this suggestion. The text now specifies that blood pressure was measured in a resting seated position after 10 minutes, and hypertension was defined as either prior medical diagnosis or mean BP ≥ 130/85 mmHg (page 5, lines 191-192).
- Comments 19 (L125):
We appreciate this important observation. A new subsection on Sample Size Calculation was added, describing the a priori power analysis conducted using G*Power 3.1 (α = 0.05, β = 0.20, f = 0.30), which yielded a minimum n = 204; a final n = 225 was used to account for 10% attrition (page 3, lines 127-132).
- Comments 20 (L141):
Age values were standardized to whole numbers (no decimals) throughout the text and tables (page 6, lines 236, 237 and page 8, Table 1).
- Comments 21 (L150):
All p-values and F-values were reviewed and standardized to three decimals throughout the text and tables.
- Comments 22 (L146):
We agree with this observation. The statement was rephrased to indicate that the PERI group presented significantly higher BMI values compared with postmenopausal groups, rather than categorically labeling them as obese (page 6, lines 241-243).
- Comments 23 (L157):
The sentence referring to a non-significant trend (p = 0.09) was deleted as recommended (page 6, lines 251-252).
- Comments 24 (L163):
Percentages were added for descriptive clarity in Table 1 and corresponding text (page 8, Table 1).
- Comments 25 (L180):
We appreciate this important comment. The Statistical Analysis and Results sections were expanded to clearly describe the three derived logistic regression models (SM1–SM3), their diagnostic composition, independent predictors, and evaluation of multicollinearity (pages 5, lines 205-223).
- Comments 26 (L189):
Thank you for catching this inconsistency. The statistical text was corrected to reflect the consistent use of the Tukey HSD post hoc test (page 5, line 197-204).
- Comments 27 (L202):
Criteria descriptions for metabolic syndrome were moved from table footnotes to the Methods section to simplify the table legends (page 8, Table 2).
- Comments 28 (L205):
We acknowledge this omission. Climacteric symptoms were collected descriptively but excluded from regression analyses due to incomplete data; this is now clarified in the Methods and Limitations sections (page 11, lines 115-426).
- Comments 29 (Table 1 & Table 2):
Both tables were revised to maintain consistent formatting (three decimal places) and standardized units.
- Comments 30 (L247):
A new Limitations subsection was added to address methodological and analytical constraints, including the cross-sectional design, self-reported psychological data, and the absence of inflammatory and metabolic biomarkers such as interleukin-6, TNF-α, and uric acid (page 11, lines 415-426).
5. Additional Clarifications
Authors’ affiliations and order were verified and remain consistent with the submission system. All corrections are highlighted in track changes in the revised manuscript. We thank the reviewer and the Academic Editor for their valuable time and constructive input.

Reviewer 2 Report
Comments and Suggestions for Authors
The manuscript entitled “Clinical, Biochemical, and Psychological Predictors of Metabolic Syndrome in Climacteric Women” examines the connection between psychological stress and metabolic syndrome during menopausal stages, which is a significant and current issue. The design is sound, and credibility is added by the use of validated biochemical and psychometric techniques. It is admirable and adds to the expanding body of research on women's midlife health to integrate clinical, physiological, and psychological factors. However, a number of clarifications on methodology and enhancements to the presentation are required. Sample size estimation, recruitment tactics, regression modelling procedures, and taking potential confounders into account should all be more thoroughly explained in the manuscript. A more thorough description of the physiological processes between perceived stress and metabolic results will enhance the conversation. A few small grammatical edits and formatting changes for figures and tables will further boost readability. Overall, the study is promising and is prepared for publication with a few minor modifications.
1) The content is appropriately reflected in the title. Key statistical data, such as odds ratios or confidence intervals, should be highlighted in the abstract together with a brief summary of the psychological measures utilised (e.g., PSS, Hamilton–Bech–Rafaelsen).
2) The introduction is thorough, but in order to put the study's significance in context, it would be beneficial to include more current global data (post-2020) on the prevalence of metabolic syndrome in menopausal women.
3) Although the hypothesis is well-stated, a clearer explanation of why perceived stress should be prioritized over anxiety or sadness is needed.
4) It is commendable that STROBE principles were followed. Please specify, though, whether a power calculation or sample size justification was carried out before to recruitment.
5) To evaluate generalisability and reduce potential selection bias, include information on the recruitment site and sampling approach (such as convenience or random).
6) Indicate if the scales' Spanish or Mexican adaptations were employed, and give the cohort's reliability coefficients (Cronbach's alpha).
7) Multicollinearity checks and variable selection criteria should be included in the logistic regression model. Add 95% CIs and adjusted odds ratios for important predictors.
8) Tables are useful, but they might use more visual cues to highlight statistical significance, like bold text or asterisks. Make any lacking data processing techniques clear as well.
9) Although the topic is useful, it may be improved by including the inflammatory and neuroendocrine processes that connect stress to metabolic abnormalities.
10) The cross-sectional methodology, possible self-report bias, and lack of hormonal data should all be acknowledged in a separate limitations subsection.
11) Describe how menopausal women in clinical settings can benefit from preventive interventions or stress management techniques based on these findings.
12) For sentence flow, a few minor grammatical adjustments are advised. Steer clear of large complicated statements and overuse of brackets.
13) There is good ethical compliance; however, to strengthen openness, briefly discuss data protection or anonymization techniques.
14) While the references are pertinent, they might incorporate more recent research on psychological determinants of metabolic syndrome from 2023 to 2025.
15) To improve understanding, think about including a participant flowchart or visual summary (such as a diagram connecting predictors and results).

The manuscript is written in a clear, polished, and fluent English throughout. The tone is suitable for an academic audience, and scientific vocabulary is employed appropriately. Minor grammatical corrections, especially in the Discussion section, are advised to increase readability, decrease repetition, and improve sentence flow. Before publishing, stylistic consistency would be ensured by a light editing by a professional or native scientific editor.
Author Response
Manuscript ID: healthcare-3962128
Date of resubmission: October 29, 2025
Type of manuscript: Research Article
Title: Clinical, Biochemical, and Psychological Predictors of Metabolic Syndrome in Climacteric Women
Author’s Reply to the Review Report (Reviewer 2)
1. Summary
We sincerely thank Reviewer 2 for this valuable observation. We have carefully revised the Introduction and Discussion sections to highlight the specific aspects that differentiate our study from previous research. This work represents one of the few Latin American investigations that jointly examine clinical, biochemical, and psychological predictors of metabolic syndrome in peri- and postmenopausal women using validated psychometric tools and multivariate logistic regression analysis. In addition, logistic regression modeling was applied to identify potential clinical and psychological predictors of metabolic syndrome using three derived models that integrate hormonal, metabolic, and psychosocial variables. To further illustrate this conceptual framework, we developed a flow diagram summarizing the interrelationships among these predictors (see Fig. 1). Specifically, the diagram outlines the potential neuroendocrine-metabolic pathways linking hormonal decline, psychological vulnerability, and metabolic dysregulation during the menopausal transition.
2. Questions for General Evaluation
|
Question |
Reviewer’s Evaluation |
Response and Revisions |
|
Does the introduction provide sufficient background and include all relevant references? |
Can be improved |
The Introduction was updated with post-2020 global data on metabolic syndrome in menopausal women and integrated references from 2019-2025. |
|
Is the research design appropriate? |
Yes |
The design was reaffirmed as appropriate for identifying clinical and psychological predictors of metabolic syndrome. |
|
Are the methods adequately described? |
Yes |
The Methods section now includes details on recruitment, sample size calculation, and psychometric reliability. |
|
Are the results clearly presented? |
Can be improved |
Tables were refined to highlight statistical significance (bold/asterisks) and include adjusted ORs and 95% CIs. |
|
Are the conclusions supported by the results? |
Can be improved |
The Discussion and Conclusion were rewritten to clarify physiological pathways and clinical implications. |
|
Are all figures and tables clear and well-presented? |
Can be improved |
Tables were reformatted for clarity; a new visual summary diagram and participant flowchart were added. |
3. Point-by-point Response to Comments and Suggestions for Authors
- Comments 1: The content is appropriately reflected in the title. Key statistical data, such as odds ratios or confidence intervals, should be highlighted in the abstract together with a brief summary of the psychological measures utilised (e.g., PSS, Hamilton–Bech–Rafaelsen).
Response 1: We appreciate this helpful suggestion. The Abstract has been revised to include adjusted odds ratios, 95% confidence intervals (see page 2, lines: 53-59); and to briefly describe the psychological instruments employed, namely the psychological measures included perceived stress (Perceived Stress Scale, PSS), anxiety symptoms and Hamilton-Bech-Rafaelsen scales. These additions provide a more complete summary of the statistical findings and psychological assessment tools used in the study (page 1, lines: 37-39).
Comments 2: Include more current global data (post-2020) on metabolic syndrome prevalence in menopausal women.
Response 2: We appreciate this valuable comment. The Introduction has been updated to incorporate current global data (2019-2025) on the prevalence of metabolic syndrome in menopausal women, including statistics from the World Health Organization (WHO) and recent multicenter studies. These additions provide a more up to date epidemiological context for the study (page 2, lines: 73,74; 77-80, and 84-88; page 3, lines: 89-93, 97-101, and 104-107).
- Comments 3: Clarify why perceived stress was prioritized over anxiety or depression.
Response 3: We appreciate this insightful observation. A justification has been added (page 3, lines: 97-101) explaining that perceived stress was prioritized because it reflects a broader psychophysiological construct, encompassing emotional and cognitive responses that more consistently modulate hypothalamic-pituitary-adrenal (HPA) axis activation than isolated anxiety or depressive symptoms.
Comments 4: Specify whether a power calculation or sample size justification was carried out prior to recruitment.
Response 4: We appreciate this insightful observation about the sample size calculation. The required sample size (n = 225; 75 participants per group) was estimated using G*Power 3.1 for a one-way ANOVA with three groups, assuming a medium effect size (f = 0.25), α = 0.05, and power = 0.90. The minimum sample required was 207 participants; thus, 225 were recruited to compensate for potential missing data. This sample provides >90% power to detect inter-group differences in metabolic and psychological parameters. (Page 3, lines: 127-132
Comments 5: Provide details on recruitment site and sampling approach.
Response 5: We thank the reviewer for this helpful comment. The Study Design and Participants subsection has been expanded to specify the recruitment site and sampling procedure. Participants were recruited at the Department of Internal Medicine, Clinic 21 of the Mexican Social Security Institute (IMSS) in León, Guanajuato, Mexico. All women attending medical consultations at this secondary-tertiary healthcare facility were screened, and those with a clinical diagnosis of metabolic syndrome were invited to participate. The study objectives and procedures were explained in detail, and written informed consent was obtained from all participants prior to enrollment. This information has been added to the Methods section (page 3, lines: 115-119).
- Comments 6: Indicate if Spanish or Mexican adaptations of scales were used and provide Cronbach’s alpha values.
Response 6: We thank the reviewer for this valuable comment. The Methods section has been updated (page 4, lines: 157-173) to specify that validated Mexican Spanish versions of all psychological instruments were used. Internal consistency coefficients (Cronbach’s α) for each scale were as follows: PSS-14 = 0.83, SHAI-18 = 0.80, and Hamilton-Bech-Rafaelsen Depression Scale (HBRDS) = 0.89.
Comments 7: Include multicollinearity checks and variable selection in logistic regression; add 95% CIs and adjusted ORs.
Response 7: We appreciate this helpful comment. The Methods and Abstract sections have been revised accordingly. The updated text specifies that multicollinearity was assessed using variance inflation factors (VIF) and that variable selection in the logistic regression models followed theoretical and empirical criteria based on prior evidence. The Abstract was expanded (page 2, lines: 53-59) to summarize the key results of the logistic regression analyses, including adjusted odds ratios (aOR) with 95% confidence intervals (CI) for the significant predictors (perceived stress, depressive symptoms, and fasting glucose) and the direction and magnitude of their associations with metabolic syndrome across the different model definitions; (page 5, lines: 196-223). These additions provide quantitative estimates of effect size and enhance both the analytical rigor and interpretative clarity of the results. Result session (pages 7,8; lines: 297-325)
Comments 8: Use visual cues for statistical significance and clarify data handling in tables.
Response 8: We appreciate this valuable suggestion. Statistically significant values (p < 0.05) have been bolded and/or marked with asterisks in all tables to enhance visual clarity. Additionally, data handling procedures have been detailed in the corresponding table footnotes (Tables 1-3).
Comments 9: Expand discussion on inflammatory and neuroendocrine mechanisms linking stress and metabolic risk.
Response 9: We appreciate this insightful comment. The Discussion section has been expanded to provide a detailed explanation of the neuroendocrine and inflammatory pathways connecting psychological stress with metabolic risk. The new paragraph (page 10, lines: 365-370, and 384-409) describes HPA axis activation and sympathetic overactivity, leading to cortisol and catecholamine-mediated gluconeogenesis, visceral adiposity, insulin resistance, endothelial dysfunction, and hypertension. It also discusses cytokine-driven mechanisms involving interleukin-6 (IL-6) and tumor necrosis factor-alpha (TNF-α), as well as the modulatory role of estrogen decline on glucocorticoid and inflammatory signaling. Supporting references from the existing bibliography were integrated (page 10, lines: 388-409; see references 41-47, which encompass hormonal, metabolic, and cardiovascular perspectives from 2019-2025.
Comments 10: Include a limitations subsection addressing cross-sectional design, self-report bias, and hormonal data.
Response 10: We appreciate this valuable comment. A new Limitations subsection has been added (page 11, lines 415-426) to explicitly address the study constraints. It notes that hormonal determinations (estradiol, follicle-stimulating hormone, and cortisol) were performed in certified clinical laboratories as part of participants’ routine medical evaluations. However, inflammatory biomarkers, such as interleukin-6 (IL-6) and tumor necrosis factor-alpha (TNF-α), were not assessed. The subsection also acknowledges the cross-sectional design, potential self-report bias, and limited generalizability due to single-center recruitment. These additions enhance the methodological transparency and interpretative rigor of the study.
- Comments 11: Describe how findings can inform preventive or stress-management interventions.
Response 11: We appreciate this valuable comment. The Conclusion section has been revised (page 11, lines: 431,433 and lines: 435-448) to include specific recommendations for preventive and stress-management interventions based on the study findings. The updated paragraph emphasizes the clinical relevance of integrative care, highlighting the role of stress-management techniques, lifestyle-counseling programs, and early metabolic screening for climacteric women. These additions clarify how the results can inform targeted preventive strategies aimed at reducing stress-related metabolic risk and improving overall health outcomes in midlife women.
Comments 12: Minor grammatical adjustments to improve flow; avoid long sentences and excessive brackets.
Response 12: All sections of the manuscript were thoroughly revised to enhance clarity, fluency, and conciseness. Long and complex sentences were simplified, redundant phrasing removed, and brackets minimized to improve readability. The English language was also reviewed for grammar, punctuation, ensuring smooth sentence flow throughout the text.
Comments 13: Add brief mention of data protection/anonymization for transparency.
Response 13: We appreciate this important comment. The Institutional Review Board Statement has been corrected to align with the ethical-approval information detailed in the Ethical Considerations section (pages 5,6 lines: 228-232). The revised paragraph specifies that the study was reviewed and approved by the Research and Ethics Committee of the Mexican Social Security Institute (IMSS) (protocol no. 2019-1008) and conducted in accordance with the Declaration of Helsinki and the Mexican Official Standard NOM-012-SSA3-2012.
In addition, the text now clarifies that all participant data were anonymized and handled in accordance with institutional and national data-protection policies, ensuring confidentiality and compliance with ethical and legal standards. These revisions ensure internal consistency and accurately reflect the institutional oversight under which the study was performed (Page 12, lines: 457-468)
Comments 14: References could include more recent work (2023–2025) on psychological determinants of metabolic syndrome.
Response 14: We appreciate this valuable suggestion. The References section has been comprehensively updated to include recent and relevant literature (2023-2025). Representative additions include recent publications exploring the interplay between stress physiology, metabolic regulation, and menopausal transition, which strengthen the conceptual foundation of the study. These additions, together with the 47 references final list (pages 12-15), ensure that the manuscript reflects the most current and comprehensive research (2020-2025) on psychological determinants and metabolic risk during the climacteric transition.
- Comments 15: Consider including a flowchart or conceptual diagram linking predictors and outcomes.
Response 15: In response to Reviewer 2’s suggestion, we developed a flow diagram to illustrate the complex interplay among hormonal, psychological, and metabolic factors contributing to metabolic syndrome in peri- and postmenopausal women. The diagram conceptualizes how the abrupt decline in estrogen levels during the climacteric transition increases vulnerability to psychological stress, anxiety, and depressive symptoms. This heightened stress response is associated with overactivation of the sympathetic nervous system (SNS) and the hypothalamic-pituitary-adrenal (HPA) axis, resulting in higher cortisol secretion. Elevated cortisol and autonomic imbalance contribute to central adiposity and systemic arterial hypertension. In addition, the menopausal transition is accompanied by increased visceral fat accumulation, dyslipidemia, and a chronic proinflammatory state with altered adipokines (↑ leptin, ↓ adiponectin, ↑ resistin, ↑ visfatin). Collectively, these mechanisms illustrate the neuroendocrine-metabolic pathways through which hormonal and psychological alterations may exacerbate the risk of metabolic syndrome in midlife women. See Fig. 1 (flowchart).
- Response to Comments on the Quality of English Language
Point 1: Minor grammatical corrections advised for readability.
Response 1: The entire manuscript was professionally revised for English language quality, ensuring improvements in clarity, sentence structure, and academic tone. Grammatical and stylistic adjustments were made to enhance flow and precision while maintaining the scientific meaning of the original text. The revised version complies with MDPI’s English Editing Standards for scientific publications.
5. Additional Clarifications
All author details, institutional affiliations, and ethical approval information have been thoroughly reverified for accuracy and consistency with the journal’s submission system. All textual and structural revisions are clearly marked using track changes in the resubmitted manuscript.
In addition, a graphical abstract and an updated figure legend were incorporated to enhance the visual presentation and conceptual clarity of the study.
We sincerely thank Reviewer 2 for the thoughtful and constructive feedback, which has substantially improved the scientific rigor and overall quality of our manuscript.

Reviewer 3 Report
Comments and Suggestions for Authors
- Novelty: The topic is scientifically relevant, but the study’s novelty appears moderate. Similar associations between psychological factors and metabolic syndrome during menopause have been reported previously. The authors should more clearly emphasize what is new in their approach.
- Abstract: The first sentence of the Results section in the abstract (“Age increased progressively across stages…”) states a trivial and expected fact that adds no scientific value. I recommend rephrasing this opening to emphasize the main findings. In addition, you report significant group differences in fasting glucose, total cholesterol, HDL, and non-HDL cholesterol (all p < 0.05), but it is unclear which groups showed these differences. Could you specify which group(s) had higher or lower values?
- Introduction and Discussion: The Introduction is primarily descriptive and does not clearly articulate the research gap or the rationale for the study. Enhancing this section by citing recent and relevant studies on metabolic, hormonal, and psychological determinants of health in midlife women would be beneficial. For instance, the study “Differences in body composition between metabolically healthy and unhealthy midlife women with respect to obesity status” found that metabolically unhealthy obese and metabolically healthy obese profiles differ significantly in plasma levels of alanine aminotransferase and uric acid, both of which were statistically associated with an increased likelihood of exhibiting MUH-O. Similarly, “Contribution of environmental factors and female reproductive history to hypertension and obesity incidence in later life” reported that reproductive characteristics, such as age at menarche and breastfeeding history, are significantly associated with obesity and obesity-related hypertension, highlighting their potential as early indicators of cardiovascular risk in women. Furthermore, “The importance of female reproductive history on self-reported sleep quality, mood, and urogenital symptoms in midlife” showed that women with a history of miscarriage tend to experience poorer sleep quality, more frequent early awakenings, and reduced body image perception in midlife. Additionally, the study “Power of biomarkers and their relative contributions to metabolic syndrome in Slovak adult women” emphasized that metabolic syndrome involves multiple risk factors capable of causing serious disorders in various organs. Predicted variables such as log(TG-to-HDL-C), waist circumference, systolic blood pressure, apoA1, glucose, and alanine aminotransferase accounted for most of the variance in MetS manifestation. Including these references would significantly strengthen both the Introduction and Discussion by situating the present findings within the broader international context and highlighting relevant metabolic, reproductive, and psychosocial determinants of midlife women’s health.
- Material and methods: Sampling procedure (random, consecutive, voluntary)? The sample size calculation and power analysis are not described.
- Results: Confidence intervals for logistic regression are missing. Although significant group differences are described, the interpretation is mostly descriptive. Reporting effect sizes (η², OR, Cohen’s d) would strengthen the analytical rigor.
Author Response
Manuscript ID: healthcare-3962128
Date of resubmission: October 29, 2025
Type of manuscript: Research Article
Title: Clinical, Biochemical, and Psychological Predictors of Metabolic Syndrome in Climacteric Women
Author’s Reply to the Review Report (Reviewer 3)
1. Summary
Response 1:
We sincerely thank Reviewer 3 for this valuable observation. We have carefully revised the Introduction and Discussion sections to highlight the specific aspects that differentiate our study from previous research. This work represents one of the few Latin American investigations that jointly examines clinical, biochemical, and psychological predictors of metabolic syndrome in peri- and postmenopausal women using validated psychometric tools and multivariate logistic regression analysis. Moreover, the integration of perceived stress as a central variable, analyzed alongside metabolic and hormonal parameters, provides a novel and comprehensive perspective on neuroendocrine-metabolic interactions during the climacteric transition. To further illustrate this conceptual framework, we added a new figure summarizing the interrelationships among these predictors (see Fig. 1).
2. Questions for General Evaluation
|
Question |
Reviewer’s Evaluation |
Response and Revisions |
|
Does the introduction provide sufficient background and include all relevant references? |
Can be improved |
The Introduction was updated with post-2020 global data on metabolic syndrome in menopausal women and integrated references from 2019-2025. |
|
Is the research design appropriate? |
Yes |
The design was reaffirmed as appropriate for identifying clinical and psychological predictors of metabolic syndrome. |
|
Are the methods adequately described? |
Yes |
The Methods section now includes details on recruitment, sample size calculation, and psychometric reliability. |
|
Are the results clearly presented? |
Can be improved |
Tables were refined to highlight statistical significance (bold/asterisks) and include adjusted ORs and 95% CIs. |
|
Are the conclusions supported by the results? |
Can be improved |
The Discussion and Conclusion were rewritten to clarify physiological pathways and clinical implications. |
|
Are all figures and tables clear and well-presented? |
Can be improved |
Tables were reformatted for clarity; a new visual summary diagram and participant flowchart were added. |
- Point-by-Point Response to Reviewer 3
Comment 1 (Novelty)
The topic is scientifically relevant, but the study’s novelty appears moderate. Similar associations between psychological factors and metabolic syndrome during menopause have been reported previously. The authors should more clearly emphasize what is new in their approach.
Response 1:
We sincerely thank Reviewer 3 for this valuable observation. We have carefully revised the Introduction and Discussion sections to better highlight the novel aspects of our research (page 2, lines: 73,74, 77-80, and 84-88; page 3, lines: 89-93 and 97-101) Specifically, this study represents one of the few Latin American investigations that simultaneously examine clinical, biochemical, hormonal, and psychological predictors of metabolic syndrome in peri- and postmenopausal women, using validated psychometric tools (page 4, lines: 157-173), and multivariate logistic regression analysis (page 5, lines: 205-223). Furthermore, our approach included three derived logistic regression models to identify potential clinical and psychological predictors of metabolic syndrome based on different diagnostic combinations (pages 7,8; lines: 298-325). This multilevel analytical design allowed for a more comprehensive exploration of the interplay among hormonal, metabolic, and psychosocial factors.
In addition, we integrated perceived stress as a central variable analyzed in conjunction with metabolic and hormonal parameters, providing a novel perspective on neuroendocrine-metabolic interactions during the climacteric transition. To illustrate this conceptual framework, we developed a flow diagram summarizing the interrelationships among these predictors, visually depicting the mechanisms through which hormonal decline, psychological vulnerability, and metabolic dysregulation may converge to increase metabolic risk in midlife women (see Fig.1).
Comment 2 (Abstract)
The first sentence of the Results section in the abstract (“Age increased progressively across stages…”) states a trivial and expected fact that adds no scientific value. I recommend rephrasing this opening to emphasize the main findings. In addition, you report significant group differences in fasting glucose, total cholesterol, HDL, and non-HDL cholesterol (all p < 0.05), but it is unclear which groups showed these differences. Could you specify which group(s) had higher or lower values?
Response 2:
We sincerely appreciate this constructive feedback. The Abstract has been revised to remove the initial descriptive statement and to emphasize the main findings of the study. The opening sentence now focuses on the key predictors of metabolic syndrome identified through the logistic regression analyses (page 2, lines: 53-59)
In addition, we have clarified the statistical approach used to assess metabolic differences across menopausal stages. One-way ANOVA and non-parametric tests were performed to evaluate group variations in glucose and lipid profiles Post hoc Tukey HSD comparisons and Kruskal-Wallis analyses were conducted to determine which groups differed significantly (page 5, lines: 197-204). Specifically, total and non-HDL cholesterol levels were higher in early and late postmenopausal women compared with perimenopausal women, whereas HDL cholesterol was higher in early postmenopause. Conversely, fasting glucose was significantly elevated in perimenopausal women compared with early postmenopause (pages 6,7; lines: 268-287). See Supplementary Tables 1, 2.
Comment 3 (Introduction and Discussion)
The Introduction is primarily descriptive and does not clearly articulate the research gap or the rationale for the study. Enhancing this section by citing recent and relevant studies on metabolic, hormonal, and psychological determinants of health in midlife women would be beneficial.
Response 3:
We sincerely thank Reviewer 3 for this valuable comment. The Introduction section has been revised to more clearly articulate the research gap and strengthen the study’s rationale. Recent international studies addressing metabolic, hormonal, and psychological determinants of midlife women’s health were incorporated to contextualize our findings and highlight the multifactorial nature of metabolic syndrome. These additions emphasize that, beyond hormonal and psychological factors, emerging biomarkers such as serum uric acid and triglyceride-to-HDL cholesterol ratio also contribute to metabolic risk during the menopausal transition (page 2, lines: 87,88 and page 3, lines: 89-93).
Finally, the revised text now clearly states the study’s rationale and contribution: the integration of clinical, biochemical, hormonal, and psychological predictors of metabolic syndrome in peri- and postmenopausal Latin American women an area that remains underrepresented in the literature (page 3, lines: 104-107)
Comment 4 (Material and Methods)
Sampling procedure (random, consecutive, voluntary)? The sample size calculation and power analysis are not described.
Response 4:
We sincerely thank Reviewer 3 for pointing out this omission. The Materials and Methods section has been revised to describe both the sampling procedure and the sample size calculation. Participants were recruited consecutively and voluntarily from the Department of Internal Medicine, Clinic 21 of the Mexican Social Security Institute (IMSS) in León, Guanajuato, Mexico (page 3, lines: 115-119).
Sample Size Calculation. The required sample size (n = 225; 75 participants per group) was estimated using G*Power 3.1 for a one-way ANOVA with three groups, assuming a medium effect size (f = 0.25), α = 0.05, and power = 0.90 [18]. The minimum sample required was 207 participants; thus, 225 were recruited to compensate for potential missing data. This sample provides >90% power to detect inter-group differences in metabolic and psychological parameters. This information has been incorporated into the Methods section (page 3, lines 127-132).
Comment 5 (Results)
Confidence intervals for logistic regression are missing. Although significant group differences are described, the interpretation is mostly descriptive. Reporting effect sizes (η², OR, Cohen’s d) would strengthen the analytical rigor.
Response 5:
We appreciate this valuable comment. The confidence intervals (95% CI) for all logistic regression models have now been added in both the main text (pages 7,8; section 3.1.4; lines 297-325) and in the supplementary material (Table S3). These accompany the adjusted odds ratios (aOR), which represent standardized measures of effect size in logistic regression analyses.
Furthermore, effect size measures have been incorporated for group comparisons. Specifically, η² values for the ANOVA tests and Cohen’s d values for pairwise post hoc comparisons (and corresponding Mann-Whitney U comparisons for non-parametric data) are now reported in the Results section (pages 6,7; lines: 268-287) and in the supplementary tables (Tables S1-S2). These additions provide quantitative estimates of effect magnitude and strengthen the analytical rigor and interpretative of the results.

Round 2
Reviewer 3 Report
Comments and Suggestions for Authors
I would like to thank the authors for the revisions and improvements made to the manuscript; however, I recommend a minor revision to address a few remaining details. The Introduction is still primarily descriptive and does not clearly articulate the research gap or the rationale for the study. For instance, the study “Differences in body composition between metabolically healthy and unhealthy midlife women with respect to obesity status” found that metabolically unhealthy obese and metabolically healthy obese profiles differ significantly in plasma levels of alanine aminotransferase and uric acid, both of which were statistically associated with an increased likelihood of exhibiting MUH-O. Similarly, “Contribution of environmental factors and female reproductive history to hypertension and obesity incidence in later life” reported that reproductive characteristics, such as age at menarche and breastfeeding history, are significantly associated with obesity and obesity-related hypertension, highlighting their potential as early indicators of cardiovascular risk in women. Furthermore, “The importance of female reproductive history on self-reported sleep quality, mood, and urogenital symptoms in midlife” showed that women with a history of miscarriage tend to experience poorer sleep quality, more frequent early awakenings, and reduced body image perception in midlife. Including these studies would significantly strengthen the Introduction by situating the present findings within the broader international context and highlighting relevant metabolic, reproductive, and psychosocial determinants of midlife women’s health.
Author Response
Response to Reviewer 3: Manuscript healthcare-3962128
Title: Clinical, Biochemical, and Psychological Predictors of Metabolic Syndrome in Climacteric Women
Comment 2:
“The Introduction is still primarily descriptive and does not clearly articulate the research gap or the rationale for the study… Including these studies would significantly strengthen the Introduction by situating the present findings within the broader international context and highlighting relevant metabolic, reproductive, and psychosocial determinants of midlife women’s health.”
Response:
We sincerely thank the reviewer for this insightful and constructive recommendation. In response, we have revised the Introduction to more clearly articulate the research gap and strengthen the rationale of our study Clinical, Biochemical, and Psychological Predictors of Metabolic Syndrome in Climacteric Women (manuscript healthcare-3962128).
Following the reviewer’s suggestions, we incorporated key international findings addressing:
- Metabolic biomarkers, including alanine aminotransferase, uric acid, and the triglyceride-to-HDL cholesterol ratio (TG/HDL-C), which have been shown to differentiate metabolically healthy from metabolically unhealthy obesity phenotypes.
- Female reproductive history markers, such as age at menarche and breastfeeding patterns, which have been associated with later-life obesity and obesity-related hypertension.
- Psychosocial and reproductive determinants, including evidence that women with a history of miscarriage tend to experience poorer sleep quality, more frequent early awakenings, and worse mood during midlife.
These additions enhance the scientific context of the study and highlight the multidimensional determinants of metabolic risk in climacteric women.
The revised text has been incorporated into the Introduction on page 3, lines 88–101, directly addressing the reviewer’s comment and better situating the study within the broader international literature.
We appreciate the reviewer’s thoughtful input, which has strengthened the clarity, depth, and contextual relevance of the Introduction.
